# Single-molecule theory of enzymatic inhibition

Tal Robin[1], Shlomi Reuveni[1,2] & Michael Urbakh[1]

The classical theory of enzymatic inhibition takes a deterministic, bulk based approach to quantitatively describe how inhibitors affect the progression of enzymatic reactions. Catalysis at the single-enzyme level is, however, inherently stochastic which could lead to strong deviations from classical predictions. To explore this, we take the single-enzyme perspective and rebuild the theory of enzymatic inhibition from the bottom up. We find that accounting for multi-conformational enzyme structure and intrinsic randomness should strongly change our view on the uncompetitive and mixed modes of inhibition. There, stochastic fluctuations at the single-enzyme level could make inhibitors act as activators; and we state—in terms of experimentally measurable quantities—a mathematical condition for the emergence of this surprising phenomenon. Our findings could explain why certain molecules that inhibit enzymatic activity when substrate concentrations are high, elicit a non-monotonic dose response when substrate concentrations are low.

---

[1] School of Chemistry and The Sackler Center for Computational Molecular and Materials Science, Tel Aviv University, 6997801 Tel Aviv, Israel. [2] Department of Systems Biology, HMS, Harvard University, 200 Longwood Avenue, Boston, MA 02115, USA. Correspondence and requests for materials should be addressed to S.R. (email: shlomire@tauex.tau.ac.il)

Enzymes spin the wheel of life by catalyzing a myriad of chemical reactions central to the growth, development, and metabolism of all living organisms[1,2]. Without enzymes, essential processes would progress so slowly that life would virtually grind to a halt; and some enzymatic reactions are so critical that inhibiting them may result in death. Enzymatic inhibitors could thus be potent poisons[3,4] but could also be used as antibiotics[5,6] and drugs to treat other forms of disease[7,8]. Inhibitors have additional commercial uses[9,10], but the fundamental principles which govern their interaction with enzymes are not always understood in full, and have yet ceased to fascinate those interested in the basic aspects of enzyme science. The canonical description of enzymatic inhibition received much exposure[1,2,11], but even at the level of bulk reactions its many limitations have already been pointed out[12]. Moreover, and despite rapid advancements in the study of uninhibited enzymatic reactions on the single-molecule level, the study of inhibited reactions has barely made progress in this direction and is still based, by and large, on what is known in bulk.

Single-molecule approaches revolutionized our understanding of enzymatic catalysis[13,14]. Early work demonstrated that at the single molecule level, enzymatic catalysis is inherently stochastic[15,16], and that one often needs to go beyond the common Markovian description to adequately account for the observed kinetics[17–20]. Universal aspects of stochastic enzyme kinetics, including the widespread applicability of the Michaelis–Menten equation and its insensitivity to microscopic details, were discovered[21–27]; and the study of enzymatic reactions under force has shed new light on the mechanics of the catalytic process[28–32].

In light of the above, it is somewhat surprising that single-molecule studies of inhibited enzymatic reactions trail behind and are just starting to emerge[33–37]. Specifically, a single-molecule theory of enzymatic inhibition, and in particular one that takes into account non-Markovian effects, is still lacking. Stochastic, single-molecule, descriptions of inhibited enzymatic catalysis can be found, but these are oftentimes based on simple kinetic schemes that fail to capture the multi-conformational nature of enzymes, or properly account for intrinsic randomness at the microscopic level. From a mathematical perspective, these kinetic schemes are usually built as Markov chains, and while one could expand them to account for increased complexity this then also compels the introduction of many additional parameters. These tend to complicate analysis, and also make it extremely difficult to discover universal principles by generalizing from simple examples. Here, we circumvent these problems by avoiding the Markov chain formulation to develop a non-Markovian theory of enzymatic inhibition at the single-enzyme level.

The simplest Markovian description of enzymatic inhibition at the single-molecule level assumes that the completion times of various processes involved in the enzymatic reaction come from exponential distributions (rates depend on process). However, and as discussed above, single-molecule experiments suggest that enzymatic catalysis is often non-Markovian. The exponential distribution should then be replaced, but the correct underlying distributions are usually unknown and guessing them is certainly no solution to this problem. Instead, we choose not to guess, allowing for catalysis, and other, times involved in the reaction to come from general, i.e., completely unspecified, distributions. This is the central and most important difference between our approach and the classical one. Rather than first, and often wrongly, assuming that all distributions are exponential (or come from some other prespecified statistics that is dictated by the structure of the Markov-chain used), and then carrying out the analysis, we show that analysis can be carried out even when underlying time distributions are treated as unknowns. Moreover, since we do not try and guess which features of the underlying distributions are important, we also do not run into the risk of being mistaken in that guess. In other words, relevant parameters emerge from our theory as output rather than being fed into it as input.

An approach similar to the one described above has previously allowed us to revisit the fundamentals of uninhibited enzymatic reactions, and show that the role of unbinding in these must be more complicated than initially perceived[38]. This then facilitated advancements in the theory of restarted first-passage-time processes[39–41] as it can be shown that the mathematical description of such processes is virtually identical to that of enzymatic catalysis at the single-molecule level. Below, we extend our approach to treat inhibited enzymatic reactions. Conclusions drawn from our analysis are then compared against conventional wisdom to predict cases where stochastic fluctuations at the level of the single enzyme would inevitably lead to a strong departure from the classically anticipated behavior.

## Results

**The classical theory of enzymatic inhibition.** The classical theory of enzymatic inhibition considers the effect of molecular inhibitors on enzymatic reactions in the bulk, and focuses on three canonical modes of inhibition (Fig. 1). In this theory, the concentrations of enzyme, substrate, inhibitor, and the various complexes formed are taken to be continuous quantities and differential equations are written to describe their evolution in time. Assuming that inhibitor molecules can bind either to the free enzyme, $E$, or the enzyme substrate complex, $ES$, as in the case of mixed inhibition (Fig. 1), and that all complexes reach fast equilibrium (the quasi-steady-state approximation), it can be shown that the per enzyme turnover rate, $k_{\text{turn}}$, of an inhibited enzymatic reaction obeys[11]

$$\frac{1}{k_{\text{turn}}} = \frac{K_{\text{m}}\left(1 + \frac{[I]}{K_{\text{EI}}}\right)}{v_{\text{max}}} \frac{1}{[S]} + \frac{\left(1 + \frac{[I]}{K_{\text{ESI}}}\right)}{v_{\text{max}}}. \quad (1)$$

Here, $[S]$ and $[I]$, respectively, denote the concentrations of substrate and inhibitor, $v_{\text{max}}$ is the maximal, per enzyme, turnover rate attained at an excess of substrate and no inhibition, and $K_{\text{m}}$ is the so-called Michaelis constant, i.e., the substrate

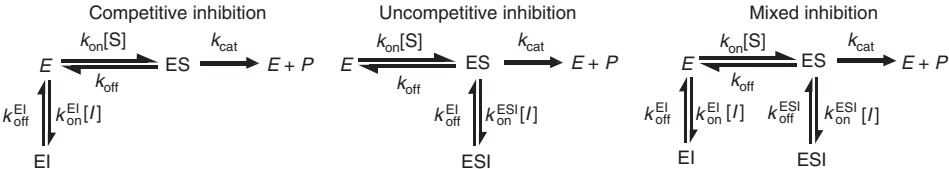

**Fig. 1** The three canonical modes of enzymatic inhibition. From left to right: competitive, uncompetitive, and mixed inhibition. Rates govern transitions between the different states: free enzyme ($E$), enzyme–substrate complex ($ES$), enzyme–inhibitor complex ($EI$), enzyme–substrate–inhibitor complex ($ESI$), and the ($E + P$) state which represents the end of a turnover cycle

concentration required for the rate of an uninhibited reaction to reach half its maximal value. Values for the kinetic parameters, $K_m$ and $v_{max}$, usually lie in the rage of $10^{-2}$–$10^6$ μm and $10^{-2}$–$10^5$ s$^{-1}$, respectively[42]; and in all subsequent examples parameters were chosen to comply with these typical values.

The parameters $K_{EI}$ and $K_{ESI}$ denote the equilibrium constants related with reversible association of the inhibitor to form the molecular complexes EI and ESI, respectively. In the classical theory, the constants, $K_m$, $K_{EI}$, and $K_{ESI}$ can all be expressed using rates of elementary processes (see Fig. 1) to get $K_m = (k_{off} + k_{cat})/k_{on}$, $K_{EI} = k_{off}^{EI}/k_{on}^{EI}$, and $K_{ESI} = k_{off}^{ESI}/k_{on}^{ESI}$, where $k_{on}$ and $k_{off}$ are the rates at which the substrate binds and unbinds the enzyme, and $k_{on}^{EI}$ $(k_{on}^{ESI})$ and $k_{off}^{EI}$ $(k_{off}^{ESI})$ are the rates at which the inhibitor binds and unbinds the enzyme (enzyme–substrate complex). Finally, note that turnover rates for the special cases of competitive and uncompetitive inhibition can be respectively deduced from Eq. (1) by taking the $K_{ESI} \rightarrow \infty$ and $K_{EI} \rightarrow \infty$ limits there.

The kinetic schemes described in Fig. 1 also serve as a starting point for a single-molecule theory of enzymatic inhibition. This theory is fundamentally different from the bulk one as it aims to describe the stochastic act of a single enzyme embedded in a "sea" of substrate and inhibitor molecules. However, the main observable here is once again the turnover rate, $k_{turn}$, which is defined as the mean number of product molecules generated by a single enzyme per unit time. Equivalently, this rate can also be defined as $k_{turn} \equiv 1/\langle T_{turn} \rangle$, where the average turnover time $\langle T_{turn} \rangle$ is simply the mean time elapsing between successive product formation events. Interpreting the kinetic schemes in Fig. 1 as Markov Chains which govern the state-to-state transitions of a single enzyme, Eq. (1) can once again be shown to hold (Supplementary Methods).

**Beyond the classical theory**. The kinetic schemes presented in Fig. (1) do not account for multiple kinetics states which are often part of the reaction. For example, it is often necessary to

discriminate between different enzyme–substrate complexes, but this could be done in a multitude of ways (Fig. 2 left) and the effect of inhibition should then be worked out on a case-by-case basis. This could work well when relevant states and transition rates can be determined experimentally, but doing so is often not possible technically or simply too laborious. Indeed, in the overwhelming majority of cases the number of kinetic intermediates and the manner in which they interconvert is simply unknown. There is thus a dire need for a description that will effectively take these intermediates into account even when information about them is partial or completely missing. Such description would also be useful when trying to generalize lessons learned from the analysis of simple case studies of enzymatic inhibition.

Generic reaction schemes could be built by retaining the same state space as in the classical approach (Fig. 1) while replacing the all so familiar transition rates with generally distributed transition times. This is done in order to account for the coarse-grained nature of states, allowing for a concise description of complex reaction schemes. The time it takes to complete a transition between two states is then characterized by a generic probability density function (PDF), e.g., $f_{T_{cat}}(t)$ for the catalysis time, $T_{cat}$, which governs the transition between the ES and $E + P$ states above (Fig. 2 right). Applied to all other transitions, an infinitely large collection of reactions schemes could then be analyzed collectively.

**Competitive inhibition at the single-enzyme level**. To concretely exemplify the approach proposed above, we consider a generic, not necessarily Markovian, scheme for competitive inhibition at the single-enzyme level (Fig. 3). As usual in this mode of inhibition, the inhibitor can bind reversibly to the enzyme to form an enzyme–inhibitor complex which in turn prevents substrate binding and product formation. However, and in contrast to the Markovian approach, here we do not assume that the catalysis time $T_{cat}$ is taken from an exponential distribution with rate $k_{cat}$, but rather let this time come from an arbitrary distribution. Since the enzyme is single but the substrate and inhibitor are present in Avogadro numbers, we assume that the binding times $T_{on}$ and $T_{on}^{EI}$ are taken from exponential distributions with rates $k_{on}[S]$ and $k_{on}^{EI}[I]$ correspondingly, but the distributions of the off times $T_{off}$ and $T_{off}^{EI}$ are once again left unspecified. We then find that the turnover rate of a single enzyme obeys (Supplementary Methods)

$$\frac{1}{k_{turn}} = \frac{K_m \left(1 + \frac{[I]}{K_{EI}}\right)}{v_{max}} \frac{1}{[S]} + \frac{1}{v_{max}}. \tag{2}$$

Note that despite the fact that it is much more general, Eq. (2) shows the exact same dependencies on the substrate and inhibitor concentrations as in the classical theory (Eq. (1) in the limit

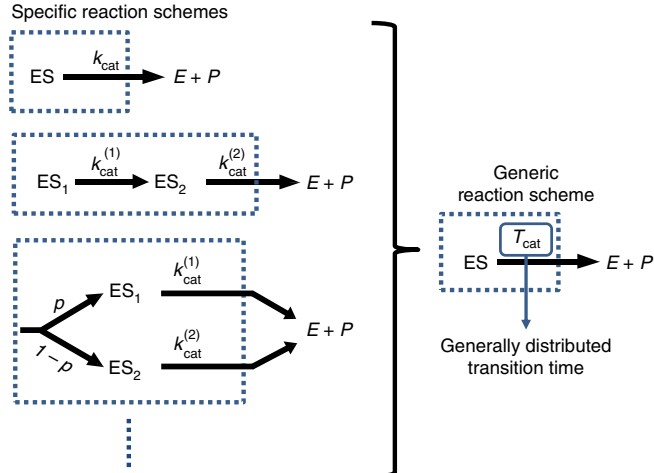

**Fig. 2** A non-Markovian reaction scheme can replace infinitely many Markovian ones. Kinetic intermediates and multiple reaction pathways could complicate the description of a reaction or various parts of it. When all intermediates and rates are known, these complications could, in principle, be addressed on a case-by-case basis. Alternatively, one could account for the non-Markovian nature of transitions between coarse-grained states by allowing for generally, rather than exponentially, distributed transition times. The main advantage of this approach is that it allows for progress to be made even when the underlying reaction schemes are not known in full, i.e., in the absence of perfect information

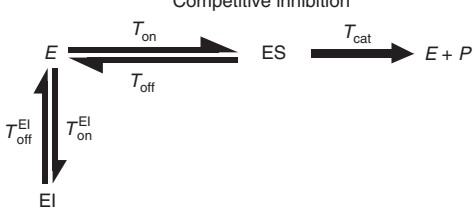

**Fig. 3** A generic scheme for competitive inhibition at the single-enzyme level. Transition rates have been replaced with generally distributed transition times to generalize the Markovian scheme in Fig. 1

$K_{ESI} \to \infty$). This result is non-trivial, and turns out to hold irrespective of the mechanisms which govern the processes of catalysis and unbinding. However, and in contrast to Eq. (1), the constants $K_{EI}$, $\nu_{max}$, and $K_m$, which enter Eq. (2), can no longer be expressed in terms of simple rates, and are rather given by (Supplementary Methods): $K_{EI} = \left(\langle T_{off}^{EI}\rangle k_{on}^{EI}\right)^{-1}$, $\nu_{max} = \Pr(T_{cat} < T_{off})/\langle W_{ES}^0 \rangle$, and $K_m = \left(k_{on}\langle W_{ES}^0\rangle\right)^{-1}$. Here, $\langle T_{off}^{EI}\rangle$ is the mean life time of the EI state (inhibitor unbinding time), $\Pr(T_{cat} < T_{off})$ is the probability that catalysis occurs prior to substrate unbinding, and $\langle W_{ES}^0\rangle = \langle \min(T_{cat}, T_{off})\rangle$ is the mean life time of the ES state (time spent in that state). Concluding, we see that while microscopic details of the reaction do enter Eq. (2), they only do so to determine various effective constants. The functional dependencies of the turnover rate on [S] and [I] are insensitive to these details, and are in this sense completely universal.

**Uncompetitive inhibition at the single-enzyme level**. We now turn to employ the same type of analysis to uncompetitive inhibition (Fig. 4). Interestingly, the situation here is very different from the competitive case analyzed above, and strong deviations from the classical behavior are observed. To show this, we follow a path similar to that taken above and obtain a generalized equation for the turnover rate of a single enzyme in the presence of uncompetitive inhibition (Supplementary Methods):

$$\frac{1}{k_{turn}} = \frac{K_m}{\nu_{max}}\frac{A([I])}{[S]} + \frac{\left(1 + \frac{[I]}{K_{ESI}}\right)B([I])}{\nu_{max}}, \quad (3)$$

where $K_{ESI} = \left(\langle T_{off}^{ESI}\rangle k_{on}^{ESI}\right)^{-1}$. Equation (3) should be compared to Eq. (1) in the limit $K_{EI} \to \infty$, and we once again see that both exhibit the same characteristic 1/[S] dependence. The dependence on inhibitor concentration is, however, different from that in Eq. (1) as Eq. (3) also includes two additional factors, $A([I])$ and $B([I])$, whose emergence is a direct result of non-Markovian stochastic fluctuations at the single-enzyme level. $A([I])$ and $B([I])$ could be understood in terms of average life times and transition probabilities (Methods), but are otherwise complicated functions of [I]. We nevertheless note that $A(0) = B(0) = 1$ always; and that in the Markovian case, i.e., when the schemes presented in Figs. 4 and 1 (middle) coincide, $A([I]) = B([I]) = 1$ for all [I]. Equation (3) then reduces to Eq. (1) in the limit $K_{EI} \to \infty$, but in all other cases analyzed this is no longer true. In particular, Eq. (3) predicts that the classical, Markovian, theory of uncompetitive inhibition will inevitably break down when catalysis times come from a non-exponential distribution.

**Breakdown of classical theory for uncompetitive inhibition**. To demonstrate the breakdown of the classical theory with a simple concrete example, we will now consider a special case of the kinetic scheme illustrated in Fig. 4. Namely, we take $f_{T_{cat}}(t) = pk_{cat}^{(1)}\exp\left(-k_{cat}^{(1)}t\right) + (1-p)k_{cat}^{(2)}\exp\left(-k_{cat}^{(2)}t\right)$, with $0 \le p \le 1$, for the PDF of the catalysis time $T_{cat}$. We thus slightly generalize the classical scheme in Fig. 1 (middle) by taking $f_{T_{cat}}(t)$ to be a mixture of two exponential densities (rather than a single exponential), but all other transitions times are still taken from exponential distributions. This form of $f_{T_{cat}}(t)$ can be shown to arise when analyzing in detail a "two-state" model where the binding of a substrate to an enzyme can occur in one of two ways, with probabilities $p$ and $(1-p)$, respectively, each leading to a different enzyme–substrate complex (ES$_1$ or ES$_2$) equipped with a distinct catalytic rate ($k_{cat}^{(1)}$ or $k_{cat}^{(2)}$). This description is equivalent

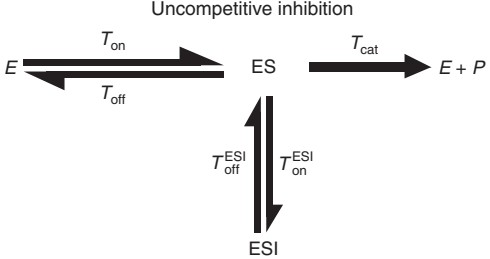

**Fig. 4** A generic scheme for uncompetitive inhibition at the single-enzyme level. Transition rates have once again been replaced with generally distributed transition times to generalize the Markovian scheme in Fig. 1

to that given here (Supplementary Methods), and we therefore refer to $f_{T_{cat}}(t)$ above as that associated with the "two state" model.

Analyzing the two-state model, we find $A([I])$ and $B([I])$ to be monotonically decreasing functions of [I] (See Fig. 5a & SI for explicit expressions). This is true as long as $k_{cat}^{(1)} \ne k_{cat}^{(2)}$ and $0 < p < 1$, and means that $A([I]), B([I]) \le 1$ for all [I]. When the inhibitor concentrations are low, these deviations from unity are linear in [I]; and for high inhibitor concentrations, both $A([I])$ and $B([I])$ eventually plateau at a certain level. Since this level could be much lower than unity, the variation in $A([I])$ and $B([I])$ may strongly affect the turnover rate in Eq. (3). Consider, for example, the limit of very high substrate concentration and note that we then have $k_{turn}^{-1}([S] \to \infty) \simeq \left(1 + \frac{[I]}{K_{ESI}}\right)B([I])/\nu_{max}$. Any deviation from the classical linear relation between $k_{turn}^{-1}$ and [I] is then because $B([I])$ is not a constant (Fig. 5b), and could thus be interpreted as a measurable telltale sign of non-Markovian kinetics.

The most important consequence of the fact that non-exponential catalysis times render $A([I])$ and $B([I])$ dependent on inhibitor concentration is perhaps the emergence of inhibitor–activator duality. This phenomenon is illustrated in Fig. 5c where we plot the turnover rate from Eq. (3) for the two-state model. The classical theory predicts that turnover should always decrease monotonically with inhibitor concentration, but here we find that this is not always the case. Specifically, we observe that for certain parameter choices (particularly when $k_{cat}^{(1)} \gg k_{cat}^{(2)}$, but also when differences between catalytic rates are not as drastic), turnover could increase with inhibitor concertation in a certain concentration range. This non-intuitive behavior is most pronounced at low-to-moderate inhibitor concentrations, and we see that at high inhibitor concentrations—where $A([I])$ and $B([I])$ are close to their asymptotic values—normal behavior is recovered (increasing inhibitor concertation lowers the turnover rate).

Our findings above demonstrate that depending on its concentration, and the inner workings of the enzyme, a molecule could act either as an inhibitor or as an activator—despite the fact that its binding always results in utter and complete shutdown of enzymatic catalysis. One way to understand this, still within the framework of the two-state model, is to realize that while the binding of such a molecule prevents product formation, it could also act as an effective switch between fast and slow catalytic states when these exist. Consider, for example, a scenario where one catalytic state is characterized by a rate that is much higher than that of the other ($k_{cat}^{(1)} \gg k_{cat}^{(2)}$). This time scale separation allows for a scenario where inhibitor binding is not frequent enough to interrupt catalysis when it proceeds through the fast catalytic pathway (hence the need for low-to-moderate inhibitor concentrations), but frequent enough so as to stop catalysis when it proceeds through the slow catalytic pathway. After the inhibitor unbinds, the enzyme could return to either of the catalytic states,

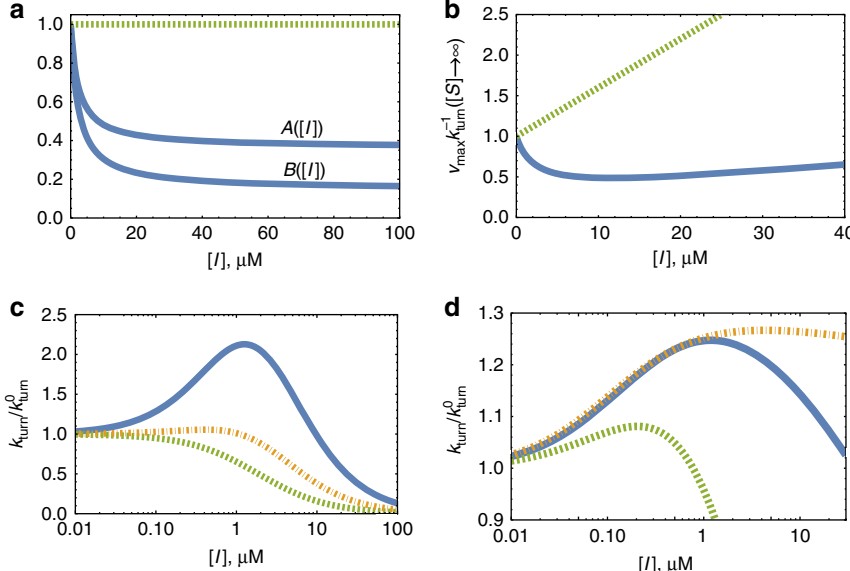

**Fig. 5** Breakdown of the classical theory for uncompetitive inhibition. **a** In solid blue, $A([I])$ and $B([I])$ from Eq. (3) for the two-state model (main text). Here, $p = 0.1$, $k_{cat}^{(1)} = 50\,\text{ms}^{-1}$, $k_{cat}^{(2)} = 0.5\,\text{ms}^{-1}$, and all other reaction times are taken from exponential distributions with $k_{off} = 2.3\,\text{ms}^{-1}$, $k_{on} = 0.1\,(\mu\text{M}\,\text{ms})^{-1}$, and $k_{on}^{ESI} = 3\,(\mu\text{M}\,\text{ms})^{-1}$. The observed behavior should be compared to that obtained for $k_{cat}^{(1)} = k_{cat}^{(2)}$ (dashed green line). The latter case coincides with the classical reaction scheme in Fig. 1 (middle), and gives $A([I]) = B([I]) = 1$ for all $[I]$. **b** The normalized inverse turnover rate $v_{max} k_{turn}^{-1}$ from Eq. (3) vs. $[I]$ in the limit of saturating substrate concentration. As in **a**, the dashed green line is drawn for the degenerate case $k_{cat}^{(1)} = k_{cat}^{(2)}$, where $B([I]) = 1$, and a linear behavior should (and is) observed. In contrast, the solid blue line is drawn for the two-state model with parameters as in **a**, and one could clearly observe strong deviations from linearity. This characteristic signature of non-Markovian kinetics is directly measurable. **c** The turnover rate (normalized by its value in the absence of inhibition) vs. $[I]$ for the two-state model with three different sets of parameters: (i) $k_{cat}^{(1)} = k_{cat}^{(2)} = 0.1\,\text{ms}^{-1}$ (dashed green); (ii) $k_{cat}^{(1)} = 10\,\text{ms}^{-1}$, $k_{cat}^{(2)} = 0.5\,\text{ms}^{-1}$ (dash-dot orange); and (iii) $k_{cat}^{(1)} = 10\,\text{ms}^{-1}$, $k_{cat}^{(2)} = 0.1\,\text{ms}^{-1}$ (solid blue). In all three cases, $p = 0.1$ and other parameters are specified in the SI. In sharp contrast to what is predicted by the classical theory, we observe that turnover may exhibit a non-monotonic dependence on inhibitor concentration. **d** The turnover rate, normalized by its value in the absence of inhibition, vs. $[I]$ for three different distributions of the catalysis time: Log-normal (dashed green), Weibull (solid blue), and Gamma (dash-dot orange), all with the same mean and variance (see Supplementary Figure 1 for details). The non-monotonic behavior of $k_{turn}/k_{turn}^0$ indicates the breakdown of the classical theory

potentially switching from slow to fast. This type of inhibitor-induced switching greatly facilitates turnover.

The emergence of inhibitor–activator duality is not unique to the two-state model, but rather a generic phenomenon whose origin we trace to stochastic fluctuations at the single-enzyme level. Depending on the enzyme, its conformations and the way they interconvert, a multitude of catalysis time distributions may arise. However, since these stem from multiple transitions between enzymatic states, the resulting catalysis time distributions would always be non-exponential and render $A([I])$ and $B([I])$ inhibitor concentration dependent. This would in turn lead to the breakdown of the classical theory. To demonstrate this, we plot the turnover rate from Eq. (3) for catalysis time distributions other than the one considered so far (Fig. 5d). In all cases, we find that within a certain concertation range the presence of an uncompetitive "inhibitor" surprisingly acts to facilitate enzymatic activity. Numerical simulations further support these conclusions (Supplementary Figure S1).

**A general criterion for inhibitor–activator duality.** The net effect resulting from the presence of an uncompetitive inhibitor also depends on substrate concentration as is demonstrated in Figs. 6a, b where we dissect the $\{[I], 1/[S]\}$ plane into three, qualitatively distinct, phases. As before, we use the two-state model to illustrate that as inhibitor concentrations increase an activator–inhibitor transition may take place. However, it can now be seen that even within this simple, two-state, toy model the manner in which the activator–inhibitor transition unfolds depends on the concentration of the substrate (Fig. 6a). Moreover, in some cases a transition does not occur at all, or only

occurs when substrate concentrations are low enough (Figs. 6b, c). Therefore, a general criterion for the emergence of inhibitor–activator duality is required.

Enzymatic reactions may involve many intermediate states and reaction pathways, and these could be different, or markedly more complex, than the two-state model we have analyzed above for illustration purposes. This bedazzling variety that enzymes display seems to hinder further progress as additional case studies usually need to be analyzed one at a time. However, the approach developed herein allows us to treat an infinite collection of reaction schemes in a joint and unified manner to determine the effect resulting from the introduction of an uncompetitive inhibitor. Analyzing the generic reaction scheme in Fig. 4, we find that a general criterion asserting the emergence of inhibitor–activator duality (i.e., asserting that $dk_{turn}/d[I]|_{[I]=0} > 0$) can be written in terms of experimentally measurable quantities (Methods). A slightly simplified version of this criterion is discussed below.

When substrate binding and unbinding are Markovian processes with rates $k_{on}[S]$ and $k_{off}$, respectively, but with inhibitor unbinding and catalysis times still allowed to come from arbitrarily distributions, we find that inhibitor–activator duality will be observed whenever (Methods, Supplementary Methods)

$$\langle T_{off}^{ESI} \rangle < \frac{1}{2}\left[CV_{W_{ES}^0}^2 - 1\right]\frac{1}{v_{max}}\left[1 + \frac{k_{off}}{k_{on}[S]}\right]. \quad (4)$$

Here, $\langle T_{off}^{ESI} \rangle$, which stands on the left-hand side for the mean life time of the ESI complex, is the only quantity in this inequality

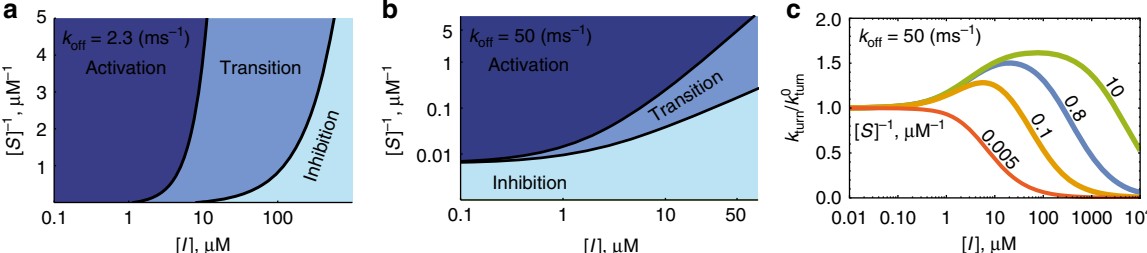

**Fig. 6** The emergence of inhibitor–activator duality depends on substrate concentration. **a**, **b** Phase diagrammatic representation of enzymatic turnover for two different instances of the two-state model. Here, activation is the phase where turnover is higher than its value in the absence of inhibition (i.e., when $[I] = 0$), and any increase in inhibitor concentration increases turnover further; transition is the phase where turnover is still higher than its value in the absence of inhibition, but where further increase in inhibitor concentration results in a decrease of the turnover rate; and inhibition is the phase where turnover is lower than its value in the absence of inhibition, and any increase in inhibitor concentration decreases turnover further still. Keeping substrate concentration fixed, and varying the concentration of the inhibitor, turnover attains a maximum when crossing the line which separates the activation and transition phases, and re-attains its value at $[I] = 0$ when crossing the line which separates the transition and inhibition phases. Plots were made with the following parameters: $k_{cat}^{(1)} = 50 \text{ ms}^{-1}$, $k_{cat}^{(2)} = 0.5 \text{ ms}^{-1}$, $p = 0.1$, $k_{on} = 0.2 (\mu M \, ms)^{-1}$, $k_{on}^{ESI} = 30 (\mu M \, ms)^{-1}$, $k_{off}^{ESI} = 50 \text{ ms}^{-1}$ and two different values of $k_{off}$ (indicated in the top left corner of each panel). **c** Lateral cross-sections through **b** showing the turnover rate, normalized by its value in the absence of inhibition, as a function of $[I]$. The activation phase in **b** corresponds to the ascending branch of the curves in **c**, whereas the transition and inhibition phases correspond to the part of the descending branch of the curves which respectively lies above, and below, unity. Substrate concentrations, corresponding to where cross-sections in **b** were taken, are indicated next to each curve

that relates to the kinetics of the inhibited enzyme. On the right-hand side, $CV_{W_{ES}^0}$ stands for the coefficient of variation (standard deviation over mean) associated with the stochastic life time of the ES complex in the absence of inhibition, and all other quantities are defined as they were immediately after Eq. 2. And so, despite the infinitely many degrees of freedom we have allowed for by leaving the distributions of inhibitor unbinding and catalysis times unspecified, predictions coming from our theory could still be tested on any enzyme of interest simply by measuring means and variances (rather than full distributions) of certain stochastic times associated with the reaction at the single enzyme level. This important feature of our theory carries over to the more general condition for the emergence of inhibitor–activator duality (Methods).

It should be noted that the criterion in Eq. (4) can only be fulfilled when $CV_{W_{ES}^0} > 1$ since $\langle T_{off}^{ESI} \rangle$ is always positive. The coefficient of variation $CV_{W_{ES}^0}$ is a dimensionless measure for the dispersion of the distribution governing the stochastic life time of the ES state in the absence of inhibition. The more dispersed (wide) this distribution is the larger $CV_{W_{ES}^0}$ and vice versa. In the classical theory, the Markovian formulation implies that time spent at the ES state is exponentially distributed (Supplementary Methods). This means that the standard deviation and mean of this time are equal, i.e., $CV_{W_{ES}^0} = 1$. In this case, and whenever the life time distribution is narrower than the exponential, inhibitor–activator duality will not be observed. Broader life time distributions with $CV_{W_{ES}^0} > 1$ are expected for enzymes with alternative kinetic pathways[19,25]; and the condition in Eq. (4) will then hold as long as substrate concentrations are low enough. Moreover, if $\langle T_{off}^{ESI} \rangle \nu_{max} < \frac{1}{2} \left[ CV_{W_{ES}^0}^2 - 1 \right]$, inhibitor–activator duality will be observed regardless of substrate concentration.

**Mixed inhibition at the single-enzyme level**. Before concluding, we note that the mixed mode of inhibition is subject to the same type of analysis applied above. In this case, we find (Supplementary Methods)

$$\frac{1}{k_{turn}} = \frac{K_m \left( 1 + \frac{[I]}{K_{EI}} \right) A([I])}{\nu_{max}} \frac{1}{[S]} + \frac{\left( 1 + \frac{[I]}{K_{ESI}} \right) B([I])}{\nu_{max}}, \quad (5)$$

and a criterion analogous to Eqs. (4) and (11) in Methods is also obtained (Supplementary Methods). In fact, all of the results in this paper could be derived by starting with Eq. (5) (Supplementary Methods), which generalizes and replaces Eq. (1) to describe enzymatic inhibition at the single-enzyme level. Specifically, note that the structure of Eq. (5), and that of Eqs. (2) and (3) as special cases, casts doubt on the ability of classical methods, e.g., that of Lineweaver and Burk[43], to reliably discriminate between different modes of enzymatic inhibition, and suggests that these methods be revised. Finally, we note that while the framework considered herein allows for arbitrary, rather than exponentially, distributed transition times between kinetic states, it still retains the common assumption (also used in the stochastic derivation of Eq. (1)) that the system "forgets" the state of origin after leaving it (In this sense, the approach presented in this paper could be said to be semi-Markovian.). Accounting for memory of past states could be important in certain cases, but the incorporation of a general form of such memory into the framework presented herein currently seems to be out of reach. Progress in this direction is an important future challenge and is anticipated to advance both theory and practice.

**Discussion**

How would the average rate at which an enzyme converts substrate into product change in the presence of a molecule whose binding to the enzyme completely shuts down its ability to catalyze? As we have shown, the answer to this question is not as simple and straightforward as it seems and curiously depends on the mode of inhibition, the molecular inner workings of the enzyme, and on a delicate interplay between substrate and inhibitor concentrations. The classical theory of inhibition provides no clue to this, but the single-enzyme approach taken herein shows that there are cases where the presence of a molecule could result in an increase of the turnover rate—even though its binding to the enzyme always results in utter and complete shutdown of enzymatic catalysis. Notably, this is not because some conformations of the enzyme–inhibitor complex are inhibitory and others excitatory or due to an interaction with some additional molecule/s[44,45] (also see "inhibition paradox"[12]), but rather because catalysis in the absence of any external modifier is non-Markovian. In other words, multiple enzyme conformations result in non-exponential transitions between coarse-grained

"states" and this, surprisingly, is already enough to produce the effect. This surprising finding not only exposes fundamental flaws in our current understanding of enzymatic inhibition, but also has direct practical implications as inhibitors are in widespread commercial use.

Take for example DAPT (N-[N-(3,5-difluorophenacetyl)-L-alanyl]-S-phenylglycine t-butyl ester), a compound tested and verified to act as an inhibitor of the enzyme γ-secretase. Developed and researched for over a decade, this once promising treatment of Alzheimer's disease was eventually abandoned as it was discovered that when administered at low concentrations, and when substrate concentrations were also low, it acted as an activator[46–49]. More awareness to the issue of inhibitor–activator duality would have surely resulted in earlier discovery of this biphasic response, saving precious time, money, and human effort.

Additional examples where inhibitor–activator duality was experimentally observed include the effects of such drugs as valinomycin, SL-verapamil, and colchicine on Pgp ATPase activity[50,51], and the effect of ADP on ATP hydrolysis by GroEL[52]. As in the case of DAPT, the qualitative nature of the phenomenon and its features are similar to those predicted by our theory, but lack of single-molecule measurements prevents us from unambiguously concluding that the mechanism we describe is indeed the one at play. Regardless, we have hereby shown that inhibitor–activator duality is inherent to the uncompetitive and mixed modes of inhibition, and that it is expected to naturally arise due to stochastic fluctuations occurring at the level of the single enzyme. Equation (4) and its generalizations then suggest that the effect, if sought for, could be observed in enzymes exhibiting multi-conformational, non-Markovian, kinetics from the kind that has already been documented in the past[17–19].

## Methods

**Definition of $A([I])$ and $B([I])$ in Eqs. (3) and (5).** $A([I])$ and $B([I])$ are defined using two auxiliary functions

$$\tilde{f}_{\text{M}}(s) = \int_0^\infty e^{-st} K_{\text{m}} k_{\text{on}} \overline{F}_{T_{\text{cat}}}(t) \overline{F}_{T_{\text{off}}}(t) \mathrm{d}t \tag{6}$$

and

$$\tilde{f}_{\text{P}}(s) = \int_0^\infty e^{-st} \frac{K_{\text{m}} k_{\text{on}}}{v_{\text{max}}} f_{T_{\text{cat}}}(t) \overline{F}_{T_{\text{off}}}(t) \mathrm{d}t. \tag{7}$$

Here, $K_{\text{m}}$, $k_{\text{on}}$, and $v_{\text{max}}$ are defined as they were right after Eq. (2) in the main text, $\overline{F}_{T_{\text{cat}}}(t) = \int_t^\infty f_{T_{\text{cat}}}(t)\mathrm{d}t$, and $\overline{F}_{T_{\text{off}}}(t) = \int_t^\infty f_{T_{\text{off}}}(t)\mathrm{d}t$. It can then be shown that (Supplementary Methods)

$$A([I]) = \frac{1 - \frac{k_{\text{on}}^{\text{ESI}}[I]}{K_{\text{m}} k_{\text{on}}} \tilde{f}_{\text{M}}\left(k_{\text{on}}^{\text{ESI}}[I]\right)}{\tilde{f}_{\text{P}}\left(k_{\text{on}}^{\text{ESI}}[I]\right)} = \frac{1 - \langle W_{\text{ES}} \rangle / \langle T_{\text{on}}^{\text{ESI}} \rangle}{\Pr\left(T_{\text{cat}} < T_{\text{off}}, T_{\text{on}}^{\text{ESI}}\right) / \Pr\left(T_{\text{cat}} < T_{\text{off}}\right)} \tag{8}$$

and

$$B([I]) = \frac{\tilde{f}_{\text{M}}\left(k_{\text{on}}^{\text{ESI}}[I]\right)}{\tilde{f}_{\text{P}}\left(k_{\text{on}}^{\text{ESI}}[I]\right)} = \frac{\langle W_{\text{ES}} \rangle \Pr\left(T_{\text{cat}} < T_{\text{off}}\right)}{\langle W_{\text{ES}}^0 \rangle \Pr\left(T_{\text{cat}} < T_{\text{off}}, T_{\text{on}}^{\text{ESI}}\right)}, \tag{9}$$

where $\langle W_{\text{ES}}^0 \rangle = \langle \min(T_{\text{cat}}, T_{\text{off}}) \rangle$ and $\langle W_{\text{ES}} \rangle = \langle \min(T_{\text{cat}}, T_{\text{off}}, T_{\text{on}}^{\text{ESI}}) \rangle$ are correspondingly the mean life times of the ES complex with and without inhibition, and $\Pr(\ldots)$ denotes the probability for the occurrence of a specified event.

**Probabilistic derivation of Eq. (4).** When will the introduction of an uncompetitive inhibitor increase the turnover rate? Consider the difference between a scenario where inhibitor molecules are not present, and a scenario where they are present at exceedingly low concentrations. Any interaction between the ES complex and an inhibitor molecule would then be very rare but will eventually happen at some point in time. In what follows, we try to determine what will be the effect this interaction has on the average time taken to compete an enzymatic reaction cycle.

An ESI complex will be formed after the inhibitor binds. It then takes the inhibitor $\langle T_{\text{off}}^{\text{ESI}} \rangle$ units of time, on average, to unbind, and for the enzyme another $\langle T_{\text{turn}}^0 \rangle - \langle T_{\text{on}} \rangle$ units of time to form a product after having just returned to the ES state. Here, the mean turnover time in the absence of inhibition, $\langle T_{\text{turn}}^0 \rangle = \frac{1}{k_{\text{turn}}^0} = \frac{K_{\text{m}}}{v_{\text{max}}} \frac{1}{[S]} + \frac{1}{v_{\text{max}}}$, was used since inhibitor concentrations were assumed to be exceedingly low. This allows us to safely neglect the probability the enzyme encounters an inhibitor again within the remaining span of the turnover cycle, and one then only needs to note that the mean substrate binding time $T_{\text{on}}$ was subtracted from $T_{\text{turn}}^0$ because the reaction continues from the ES state rather than starts completely anew. In total, a product will then be formed, on average, after $\langle T_{\text{remain}}^1 \rangle = \langle T_{\text{off}}^{\text{ESI}} \rangle + \langle T_{\text{turn}}^0 \rangle - \langle T_{\text{on}} \rangle$ units of time.

Suppose now that instead of having the inhibitor bind the ES complex as described above, the reaction would have simply carried on uninterruptedly from that point onward, i.e., as it would in the absence of inhibition. How much time would it then take it to complete? To answer this, we observe that the inhibitor encountered the ES complex at a random point in time, as opposed to immediately after its formation. Having already spent some amount of time at the ES state, the mean time remaining before the system exits this state need not necessarily be identical to the mean life time, $\langle W_{\text{ES}}^0 \rangle$, of a freshly formed ES complex in the absence of inhibition. Indeed, the time we require here is the mean residual life time of the ES complex, i.e., starting from the random point in time at which it encountered the inhibitor and onward. A key result in renewal theory then asserts that, when averaged over all possible encounter times, the mean residual life time is given by $\frac{1}{2} \langle W_{\text{ES}}^0 \rangle + \frac{1}{2} \frac{\sigma^2\left(W_{\text{ES}}^0\right)}{\langle W_{\text{ES}}^0 \rangle}$[53], where $\sigma^2\left(W_{\text{ES}}^0\right)$ denotes the variance in $W_{\text{ES}}^0$. This time could be larger, or smaller, than the mean life time $\langle W_{\text{ES}}^0 \rangle$, and the two are equal only when $\sigma^2\left(W_{\text{ES}}^0\right) = \langle W_{\text{ES}}^0 \rangle^2$—as happens, for example, in the case of the exponential distribution.

After the system exits the ES state two things could happen. If a product is formed the reaction there ends. Otherwise, the enzyme reverts back to its free state, and the reaction takes, on average, another $\langle T_{\text{turn}}^0 \rangle$ units of time to complete. When the enzyme first enters the ES state the probability that a product is formed is $\Pr(T_{\text{cat}} < T_{\text{off}})$. What is, however, the probability that a product is formed from an ES complex that is first observed at some random point in time as in the scenario described above? Looking at the total time an enzyme spends at the ES state across many turnover cycles, this probability should coincide with the relative time fraction taken by ES visits which end in product formation, and this is given by $\Pr(T_{\text{cat}} < T_{\text{off}}) \langle W_{\text{ES}}^0 | T_{\text{cat}} < T_{\text{off}} \rangle / \langle W_{\text{ES}}^0 \rangle = \Pr(T_{\text{cat}} < T_{\text{off}}) \langle W_{\text{ES}}^0 | \text{ES} \rightarrow E + P \rangle / \langle W_{\text{ES}}^0 \rangle$, with $\langle W_{\text{ES}}^0 | \text{ES} \rightarrow E + P \rangle$ standing for the average time spent at the ES state given that a product was formed thereafter. Summing the contributions above, we see that when the reaction is left to proceed in an uninterrupted manner a product will be formed, on average, after $\langle T_{\text{remain}}^0 \rangle = \frac{1}{2} \langle W_{\text{ES}}^0 \rangle + \frac{1}{2} \frac{\sigma^2\left(W_{\text{ES}}^0\right)}{\langle W_{\text{ES}}^0 \rangle} + \langle T_{\text{turn}}^0 \rangle \left(1 - \Pr(T_{\text{cat}} < T_{\text{off}}) \langle W_{\text{ES}}^0 | \text{ES} \rightarrow E + P \rangle / \langle W_{\text{ES}}^0 \rangle \right)$ units of time.

Concluding, we observe that for the introduction of an inhibitor to facilitate turnover one must have $\langle T_{\text{remain}}^0 \rangle > \langle T_{\text{remain}}^1 \rangle$, or equivalently

$$\langle T_{\text{off}}^{\text{ESI}} \rangle < \langle T_{\text{on}} \rangle + \frac{\langle W_{\text{ES}}^0 \rangle}{2} \left[ 1 + \frac{\sigma^2\left(W_{\text{ES}}^0\right)}{\langle W_{\text{ES}}^0 \rangle^2} \right]$$
$$- \langle T_{\text{turn}}^0 \rangle \frac{\Pr(T_{\text{cat}} < T_{\text{off}}) \langle W_{\text{ES}}^0 | \text{ES} \rightarrow E + P \rangle}{\langle W_{\text{ES}}^0 \rangle}. \tag{10}$$

Recalling that $\langle T_{\text{on}} \rangle = (k_{\text{on}}[S])^{-1}$, $v_{\text{max}} = \Pr(T_{\text{cat}} < T_{\text{off}}) / \langle W_{\text{ES}}^0 \rangle$, and $K_{\text{m}} = \left(k_{\text{on}} \langle W_{\text{ES}}^0 \rangle\right)^{-1}$, the condition in Eq. (10) (emergence of inhibitor–activator duality) can be rearranged and shown equivalent to

$$\underbrace{\frac{\langle T_{\text{off}}^{\text{ESI}} \rangle}{\langle W_{\text{ES}}^0 \rangle}}_{\text{ratio of mean life times}} < \underbrace{\frac{1}{2}\left[ CV_{W_{\text{ES}}^0}^2 - 1 \right]}_{\text{contribution from statistical fluctuations in life time of ES complex}}$$
$$+ \underbrace{\left[ 1 - \frac{\langle W_{\text{ES}}^0 | \text{ES} \rightarrow E + P \rangle}{\langle W_{\text{ES}}^0 \rangle} \right]}_{\text{contribution from bias in breakdown of ES complex}} \Big/ \underbrace{\frac{[S]}{k_{\text{m}} + [S]}}_{\text{fraction of time spent at ES (no inhibition)}}. \tag{11}$$

Once again, we turn attention to the fact that the condition in Eq. (11) only involves means and variances (rather than full distributions or higher moments) of stochastic times associated with the reaction at the single-molecule level, and that all terms in this equation are experimentally measurable. Equation (4) in the main text follows from Eq. (11) by further assuming that the time for substrate unbinding is exponentially distributed with rate $k_{\text{off}}$ (Supplementary Methods). An alternative derivation of Eq. (11) is given in Supplementary Methods.

**Data availability**. Data supporting the findings of this manuscript are available from the corresponding author upon reasonable request.

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

## Acknowledgements

We gratefully acknowledge support from the German-Israeli Project Cooperation Program (DIP). S.R. gratefully acknowledges support from the James S. McDonnell Foundation and the Azrieli Foundation; and would also like to thank Johan Paulsson and the Department of Systems Biology at Harvard Medical School for their support and hospitality.

## Author contributions

T.R., S.R., and M.U. conceived the work, derived the results, and wrote the paper.

## Additional information

**Competing interests:** The authors declare no competing financial interests.

