## [Peer Review File · Nature Communications]

Editorial Note: This manuscript has been previously reviewed at another journal that is not operating a transparent peer review scheme. This document only contains reviewer comments and rebuttal letters for versions considered at Nature Communications. Mentions of prior referee reports have been redacted.

Reviewers' Comments:

Reviewer #1:

Remarks to the Author:

In the present manuscript, Robin et al. present a theoretical analysis of enzymatic inhibition based on single-molecule theory, and argue that this theoretical framework predicts the emergence of inhibitor-activator duality. This work is primarily conceptual, although indirect links are made to some example enzymes on pg. 11 of the main text.

While this analysis may be suitable for a specialist physics or mathematically-oriented journal, I do not think it is suitable for Nature Communications.

1. The biggest caveat in the manuscript is the lack of either experimental support or more detailed computational models. The fundamental idea of single molecule analysis similar to Michaelis-Menten is already in the literature, and is not in and of itself novel. There has, for example, been extremely elegant work by most notably Xie and English, and this work does not make a substantial contribution beyond what is already in the literature (while there are novel aspects, these are incremental in nature compared to the published work).

2. Following from my first concern, I do not believe the paper demonstrates how the analysis presented by the authors provides unique insight into real biochemical or biological systems. Some examples are presented on pg. 11, but this is indirect, and if the authors' framework were broadly applicable, real examples should be presented accompanied by appropriate quantitative analysis.

These are major concerns, and in light of these, I believe the manuscript is more suitable for a specialist discipline-specific journal and is not suitable for the broad readership of Nature Communications.

Reviewer #2:

Remarks to the Author:

I appreciate the efforts invested by the authors into the rebuttal and the revised manuscript. In particular I would second the authors in their rebuttal of Referee B: rate equations are not always the fundamental start. On the contrary, we are discovering more and more complex systems, in which the rate equation approach needs to be replaced by more elaborate schemes. In some sense, this has been underlined already by the ligand rebinding experiments of Hans Frauenfelder which showed extended power-law rebinding distributions. In the revision process several points could be clarified, and I am absolutely convinced that this work should be published in a good outlet.

Now to the important question: should this work be published in Nature Communications? I did express my initial reservations. However, in this revised form I am inclined to follow the authors' line of argument. This is potentially very relevant material. It is hard to invade a classical domain such as enzymatic biochemical reactions, and this will certainly be facilitated when theoretical results are promoted at a very high level. Therefore I do warmly support publication of this revised manuscript in Nature Communications.

Reviewer A

Reviewer A Wrote

“In the present manuscript, Robin et al. present a theoretical analysis of enzymatic inhibition based on single-molecule theory, and argue that this theoretical framework predicts the emergence of inhibitor-activator duality. This work is primarily conceptual, although indirect links are made to some example enzymes on pg. 11 of the main text. While this analysis may be suitable for a specialist physics or mathematically-oriented journal, I do not think it is suitable for Nature Communications.”

We are sorry that reviewer A feels that our paper is more suitable for publication in a specialized journal, but disagree with his/her opinion. Our stand, explaining why it is imperative that the paper be published in a journal directed at the wide scientific community, was elaborately discussed in the previous letter of reply and will not be duplicated here.

Reviewer A Wrote

“1. The biggest caveat in the manuscript is the lack of either experimental support or more detailed computational models. The fundamental idea of single molecule analysis similar to Michaelis-Menten is already in the literature, and is not in and of itself novel. There has, for example, been extremely elegant work by most notably Xie and English, and this work does not make a substantial contribution beyond what is already in the literature (while there are novel aspects, these are incremental in nature compared to the published work).

2. Following from my first concern, I do not believe the paper demonstrates how the analysis presented by the authors provides unique insight into real biochemical or biological systems. Some examples are presented on pg. 11, but this is indirect, and if the authors' framework were broadly applicable, real examples should be presented accompanied by appropriate quantitative analysis.”

Reviewer A acknowledges that there are novel aspects in our work, but it is hard for us to understand how he then continues to argue that these are incremental in nature. The reason for this is simple. The pioneering work of Xie and English provided support for the existence of non-Markovian effects in enzymatic reactions, but the broader implications of these were not fully understood at the time. The main contribution of our work in that regard is to show that such non-Markovian effects could surprisingly make inhibitors act as activators. This means that the classical theory of enzymatic inhibition needs to be replaced with a new theory which we develop in the paper. One may then argue that the new theory is somehow flawed (we stand behind every equation), but arguing that all it offers is

incremental advancement on top of existing knowledge does not fall in line with the fact that the new theory provides counterintuitive predictions that are *qualitatively* different than those offered by the classical one. Put differently, this study rejects the existing approach and offers to replace it with another.

As for the link with more detailed computational models, here too we must say that we do not understand the reviewer's stand. The results presented in our paper were derived analytically and cross validated numerically. Moreover, in our previous letter of reply we invited reviewers to ask for additional numerical validations as we were positive that the result of these would further support our conclusions.

Regarding connection with experiments, in the revised version of the manuscript we have made a serious effort to extend discussion on available experimental evidence that support our findings. However, we kindly ask the reviewer to keep in mind that this work is theoretical and that conclusive experimental support for our findings can only come from cutting-edge single-molecule experiments that are yet to be performed. Our predictions also bare implications on bulk behavior, but making an unambiguous connection between the emergence of inhibitor-activator duality in bulk and the mechanism proposed in our theory would also require novel single-molecule experiments. The main contribution of our work in this regard is that it could motivate people to go on and do experiments they would otherwise not attempt. Together with the theory formulated herein, such experiments could drastically change the way we think about enzymatic reactions.

Reviewer B

Reviewer B Wrote

"I appreciate the efforts invested by the authors into the rebuttal and the revised manuscript. In particular I would second the authors in their rebuttal of Referee B: rate equations are not always the fundamental start. On the contrary, we are discovering more and more complex systems, in which the rate equation approach needs to be replaced by more elaborate schemes. In some sense, this has been underlined already by the ligand rebinding experiments of Hans Frauenfelder which showed extended power-law rebinding distributions. In the revision process several points could be a good outlet. Now to the important question: should this work be published in Nature Communications? I did express my initial reservations. However, in this revised form I am inclined to follow the authors' line of argument. This is potentially very relevant material. It is hard to invade a classical domain such as enzymatic biochemical reactions, and this will certainly be facilitated when theoretical results are promoted at a very high level. Therefore, I do warmly support publication of this revised manuscript in Nature Communications."

We would like to thank the Reviewer for his/her support. Indeed, rate equations carry with them a set of restrictive assumptions which are often not suitable for the analysis of complex systems such as enzymes. Experimental observations indicating that the description of enzymatic catalysis should no longer rely on rate equations are now widespread, but adoption of more sophisticated approaches is currently hindered due to historical and political reasons. We thank the reviewer for seeing this clearly and for supporting the dissemination of our work to the broad readership of Nature Communications